# Creative Narration as a Design Technique

**Dimosthenis Manias * and Irene Mavrommati**

School of Applied Arts and Sustainable Design, Hellenic Open University, 26335 Patras, Greece;
mavrommati@eap.gr
* Correspondence: dimosthenis.manias@gmaill.com

**Abstract:** Creative narration is a structured ideation technique based on storytelling. It has the potential to enhance the initial design process of ideation in terms of collaboration and creativity. People from various disciplines, following specific steps, collaborate to create a story. Afterward, inspired by their stories, they create products and services. In this paper, two case studies are presented and compared, where the technique of creative narration was used in the contexts of two creative workshops. An initial assessment of this process, highlighting the strong and weak points of the technique, is discussed in this paper.

**Keywords:** design automation; creative narration; technical evaluation; ideation; design thinking



## 1. Introduction

The Fourth Industrial Revolution presupposes the seamless coexistence of man and technology, and their co-evolution in ways presently unforeseen. Engineers need to include creativity and imagination, to grasp ideas of a future that "can be made" possible and to be more flexible in the way they perceive the problem space. They need to practice broad ideation skills, think divergently as well as convergently, and be able to use techniques to facilitate thinking outside the expected norms ("outside the box"). Appropriate ideation design processes should be capable of adapting to a dynamic and evolving world; leave room for people to express themselves; and be structured, understandable and effective.

Current innovation processes are not completely reliable in terms of innovation production (Kumar, 2012) [1], although they become increasingly relevant, as in the last two decades, the market has shifted from price-based competition to knowledge-based competition (Nyholm, & Langkilde, 2003) [2].

Avouris et al. (2018) [3] stated that the engineering students tend to deal with technologies they feel familiar with. The produced scenarios often do not imply innovation, do not solve the identified problems in novel ways. Namely, a thought is considered "inside the box" because it fails to create non-functional requirements or does not answer bigger questions.

The suggested ideation technique introduced in this research, tries to improve the scenario-creating process by providing, in a structured way, possibilities for conceiving a world and its main characters within scenarios, and by helping the designers to follow these initial assumptions during the whole creative process. It forces them to find solutions to imaginary problems in an intelligible way. However, it is not only a scenario technique. The designers are transformed into writers, and as such, they become creative and immersed in the worlds they create. The dynamics of true story creation connect them to a common goal instead of separating them into their worldviews.

In this paper, two workshops deploying techniques related to creative narration are presented. The first one was an introduction to the technical workshop where the users' first impressions and opinions were gathered. The second workshop was tailored to compare and investigate the use of another related technique, impact mapping. Impact mapping is an already established technique based on a more strict structure that has common

characteristics with creative narration. The common characteristics are the actors, the scope and the deliverables, which are the elements encoded inside a story as well. Since impact mapping is a more strict technique, the second workshop aimed to investigate the spectrum between a creative, abstract technique and a structured design technique. Three classrooms were created, one with creative narration, the second with a blend between creative narration and impact mapping and the third with impact mapping alone.

In the first part of the paper, the theoretical pillars, on which the creative narration technique is based, are presented, followed by the related work. In the next part, the two workshops are presented, together with their results. The paper finishes with the conclusions and the discussion for future research.

### 1.1. Current Situation

The idea generation process in design is a creative process that borrows several techniques from the arts, including literature, theater, visual arts, cinematic arts, etc. During the first stage of the design process, ideation, various disciplines participate and contribute to the functional and non-functional specifications of the designed product. Innovation and creativity are some of the teams' concerns. Design processes aim to discover the users' needs, explore the context of use, determine the interactions between all involved actors and visualize the future (Nyholm, & Langkilde, 2003) [2].

#### 1.1.1. Creativity

There are two theories in Western culture about creativity. The idealistic theory supports that ideas are created spontaneously by the subject's subconscious, and the theory of action, which claims that creation is the result of a process. Scientists have proved that the theory of action is closer to reality (Sawyer, 2011) [4]. Innovation can be achieved through a defined procedure, following several phases. "Balloon" is the simplest model for creativity and it has two stages In the first stage, many ideas are created through "divergent thinking," and in the second stage they all end up in one through "convergent thinking" (Sawyer, 2011) [4].

Creative problem solving (CPS) is considered an established method for identifying a problem or a challenge and then dealing with it imaginatively and innovatively. It helps redefine the problems and opportunities in coming up with new, innovative answers in order to solve them and then take action (Osborn, 2012) [5].

The steps of the creative process as mentioned in the CPS model (Osborn, 2012) [5] are:

1. Orientation: clarify and identify the problem.
2. Preparation: collect the relevant information.
3. Analysis: analyze the collected data.
4. Hypothesis: creation of alternative solutions.
5. Incubation: pause and wait for new ideas.
6. Synthesis: gather all the different pieces.
7. Evaluation: test the results and new solutions and products.

#### 1.1.2. Design Thinking

Design thinking started in the business sector as a process for the development of new products. The term refers to techniques that provide practical solutions to problems that require creativity. Such problems are the ones that are complex, vaguely defined or even unknown. In design thinking, problems are addressed by techniques that designers use when designing (Brown, 2008) [6]. Owen (2007) [7] argues that design thinking, in contrast to traditional organizational and management techniques, delays decision making to maximize learning as a strategy to reduce uncertainty. Learning is essential to design (Beckman and Barry, 2007) [8].

In his work, Bratitsis (2018) [9] compared the creative process of creating a digital story with the process of design thinking. This comparison is similar to that of Maiden et al. (2010) [10], who contrasted the Osborne(–Parnes) (1953) creative problem solving model

with software development processes [11]. Both studies contrasted a purely creative process and a design process. The two models have similarities, but they also have differences. What is important from the comparison is that designing bears similarities to a creative process.

The five steps of design thinking are:

1.  Empathize: In this step, an empathetic understanding of the problem is attempted, usually through user research.
2.  Define: Here the data from the previous stage is synthesized to identify the core problems that are formulated into problem statements. In this step, persons can be utilized to keep the process human-centered.
3.  Ideate: In this step, ideas are generated, and brainstorming is a great tool for this end. The knowledge that emerged from the two previous phases opens the door for the designer to think out of the box and find new solutions to the problem.
4.  Prototype: This is the phase for experimentation by creating and testing the inexpensive implementation to investigate the ideas they have generated.
5.  Test: In this phase, the prototypes are tested. Although this is the final phase, the results are used to redefine and rework the previous phases. Design thinking is an iterative process.

*1.2. Design Tools Categories*

There are three major categories of product design and development techniques. The "traditional," the "design thinking" and the "lead user" ones. The "design thinking" and the "lead user" development techniques support user-driven innovation, and they are the results not only of the academic research but also of the experience of Tim Brown, Eric von Hippel and David Kelley. The difference between the "voice of the consumer (design thinking) methods" and the "lead user methods" is that in the first type the manufacturers focus on their efforts toward identifying the needs of the user, whereas in the second type they cooperate with lead users and they make the products together.

This research focuses on the design thinking method, which is considered a modern, user-centered, creative method. Some creative tools that are used in this method are the following:

*   Role playing (Vaajakallio and Mattelmäki, 2014) [12].
*   Games (Vaajakallio and Mattelmäki, 2014) [12].
*   Card tools (Mora, et al., 2017) [13].
*   Scenarios (Carroll, 2000) [14].
*   Storytelling (Kim, 2019) [15], (Quesenbery and Brooks, 2010) [16].

Authentic creative problem-solving methods such as CPS and Synectics were created, along with several creative techniques that can be used and expanded. Even with a simple search, someone could find hundreds of different techniques that have generated innovative ideas.

*1.3. Scenario-Based Design*

The scripts constructed in the scenario-based design process called scenario-based design (SBD) generally do not follow any theory of dramatic structure. It is not required, for example, to have a beginning, a middle and an end. In creative storytelling on the other hand, the scripts follow specific techniques to produce better scenarios and to let participants cooperate more efficiently.

Scenarios are stories—stories about people and their activities. The scenarios emphasize goals that are derived from the structure of the system. Scenarios consist of distinctive elements (Propp, 1958) [17]. They either refer to scenery or imply scenery. e.g., for an accountant's scenario, the scenery would be a computer with a computer spreadsheet and the accountant sitting at his desk in front of a computer. The scenario would also refer to the location of the spreadsheet on the computer screen with an open folder behind it. Scenarios include actors. In the above example, the only actor is the accountant. However, in human

activities, usually multiple people with different goals act together. Each scenario has a plot and includes at least one actor with a goal. Scenarios are the minimal environments in which user-centered design is practiced.

Creating scenarios pushes designers to go beyond default responses. Being compact but also flexible helps with managing vague and dynamic situations. Scenarios are considered compact because they confront one design aspect while they provide a specific solution. They are flexible because they are intentionally open and easy to review, dismiss or develop further.

## 2. Materials and Methods

### 2.1. Workshops Teaching Tools

Google Classroom and Zoom are the teaching tools that were used. Google Classroom is a free service developed and offered by Google that serves as a virtual class, where users can share content and create and grade assignments. It consists of two types of users, students and teachers. The workshop described the whole structure; the content and the assignments were provided through a Google Classroom, where students had the opportunity to submit their work with the use of Google Docs, since the classroom collaborated on the software level and service level with the Google Docs suite.

Zoom is a cloud-based service that provides videotelephony and online chat services. For the second assignment (hero's journey), students were given instructions and then were divided into two-person breakout rooms. The moderator of the workshop visited each breakout room to provide clarifications and guidance.

### 2.2. Creative Narration as an Ideation Technique

#### 2.2.1. Scope

The research reported here explores the use of creative narration in terms of ease of use and effectiveness. To meet this aim, the author developed a specific technique, borrowing from narrative techniques, and defined its steps in detail, so that it would be easy to explain and practice. Then, during a creative ideation workshop, six engineering students used the technique, moderated by an instructor, and then evaluated it.

#### 2.2.2. Creative Narration Technique

The creative narration technique consists of three parts. In the first one, participants are asked to practice free writing. This is a warm-up that encourages participants to let themselves freely write whatever they want. Furthermore, it allows them to operate with their subconscious skills. The Section 2 encourages participants to create a scenario using the steps of Joseph Campbell's Hero Journey (Campbell, (2008) [18]. This detailed technique guides participants to a well-rounded story that does not miss any important elements. The Section 3, with a step-by-step approach, aids participants in creating a poem. In the beginning, it asks them to remember a lyric, then to create a simile and a metaphor and lastly to write the poem. They are free to write a Japanese Haiku or a Cretan Mantinada in case they feel more comfortable with that.

#### 2.2.3. Ideation

Ideation is an inspiration method that encourages designers to bring about new ideas at conferences (e.g., brainstorming or worst possible idea). It constitutes the third phase in the design thinking process. During the ideation period, all the participants get together, and through free thinking, they try to develop a great number of ideas in a facilitated, judgment-free environment.

The purpose of the ideation phase is to produce solutions using creativity and innovation. Extending the solution field provides the ability to the design team to search further than the typical methods of solving problems, to discover superior, more elegant and fulfilling answers to problems that involve in the user's experience of a product (Dam, & Siang, 2021) [19].

During the design thinking process, the Ideation phase frequently comes along with the first two parts; the first one is the "empathize" part and the second is the "define" part. There is an important overlap between the "define" and "ideation" phases in a common design thinking process. This overlap can be found in the kinds of methods design teams employ during these two phases. For example, bodystorm and "How Might We" questions are often used in both of these stages.

### 2.2.4. Creative Narration: Rationale

Kim and Lee (2019) [15], in their work on service design and storytelling, explore the narrative models that can be used in service design matching each model with a design stage. Their work covers the analysis of the narrative models in terms of design, and the next step is to create a design model that uses those narrative models. The book by Quesenbery, W. and Brooks, K. (2010) [16] discusses the benefits of using stories in service user-experience design. Stories are used by designers to explain and activate the imagination, and better understand the users. An important part of the evaluation tests where stories can be used for usage scenarios, but also as fictional scenarios to leave the user more freedom to express himself. The latter is more widely used in areas such as Asia, where users cannot easily express negative criticism. Another use is to capture the needs of the user because, instead of extensive descriptions of their needs, users can describe them by making a very short story with a specific structure.

Some tools that are used in creative writing and might be useful to service design according to (Kim & Lee, 2019) [15] are the semiotic square, Propp's 31 functions (Propp, 1958), the Triangle of Mimetic Desire (Girard, & Freccero, 1965) [20], the hero's journey (Campbell, 2008) [18] and the Freytag model (Freytag, 1894) [21]. In Snævarr, S. (2018) [22], poems are mentioned as the best vehicle for writers to express their feelings and their subconscious inklings. This is the reason why they were selected as a part of the technique tested in this research. The other one was the hero's journey, a tool with very well-described steps that allow the writer to sketch a whole story, and the first tool was free writing, which is a warm-up tool that can make the writers open up and connect their hands with their hearts and their minds.

In 2022, while this research was in progress, another similar study came to light. Fletcher and Benveniste (2022) [23] researched how training with narrative theory would enhance creativity. The similarity with the current study is that narrative theory is utilized to stimulate creativity; the difference from the technique that is presented here is that creative narration is a technique tailored to produce creativity and not a training tool.

### 2.3. Impact Mapping

Impact mapping was firstly presented by Gojko Adzic in his book, "IMPACT MAPPING Making a big impact with software products and projects," in 2012. Impact mapping stands as a strategic planning technique whose main role is to help organizations not get lost during their process of developing and launching new projects. What impact mapping gives to different teams is the ability to synchronize their actions with inclusive business purposes and make better roadmap decisions. Not all products and projects that come up are meant to work in a vacuum. Their purpose is to interact with people, other projects, the main organization and the wider community around them. While all the well-known planning methods assume that the world will remain the same, it does not, and they do not dare to visualize any long-term projects which are followed by a significant communication gap between business sponsors and delivery teams. Therefore, impact maps can picture the string relationship between delivery plans and the rest of the world, capturing the most important assumptions and delivery scope. With them, it is easier and more effective to react to change, while still providing a good road map for delivery teams and a big-picture view for business sponsors. More importantly, impact mapping supports waste reduction by avoiding scope creep and over-engineered solutions. It also offers a focus on delivery by

putting deliverables in the context of the impacts they are supposed to achieve. It enhances collaboration by creating a shared big-picture view for technical and business people too.

Impact mapping has several unique advantages over similar methods:

1.  It is based on a method invented by an interaction design agency and similar to a team-building method, which means that it facilitates collaboration and interaction. It is significantly less bureaucratic and much easier to apply than many alternatives. It also facilitates the participation of groups of people from different backgrounds, including technical delivery experts and business users, helping organizations use the wisdom of crowds.
2.  It visualizes assumptions. Alternative models mostly do not communicate assumptions clearly. Impact mapping does, and because of that, it helps teams to make better decisions in rapidly changing environments, such as IT. The visual nature of this method also facilitates effective meetings and supports big-picture thinking, which provides organizational alignment.
3.  It is fast. It fits nicely with iterative delivery models that are now becoming mainstream in software.

Very few people working on delivery know the expected business objectives. These are sometimes drafted in a vision document, but more frequently exist only at the back of senior stakeholders' minds. Even when they are communicated, business goals are often defined in vague terms. Knowing why we are doing something is the key to making good decisions about cost, scope and timelines, both at the start and later when things change. Good goals tend to be SMART: specific, measurable, action-oriented, realistic and timely. To make all the steps of an impact map and show their actual purposes, it is necessary to have "actors" incarnate them. To deliver high-quality results, we first have to understand who these people are, and what kind of value they are looking for from our products or project outcomes. In addition to those directly getting value out of our software, we also have to consider a host of others who can make decisions that influence the success of a product milestone or the outcome of a project. The software does not work in a vacuum, and it rarely controls all the actors who are involved with it. People have their own needs, goals and preferences, which all come into play if we truly care about achieving a business goal instead of just delivering software. Impact maps make us think about all these decision makers, user groups and customer segments. By mapping out different actors, we can prioritize work better—for example, focusing on satisfying the most important actors first.

The second level of an impact map sets the actors from the perspective of our business goal. By listing impacts on the second level of a map, we consider the desired changes in the behavior of actors. This leads to better plans and helps with prioritization. Different actors could help us or obstruct us in many ways on our route to achieving the key business objectives. Some of the impacts will be competing, some conflicting and some complementary. We do not necessarily have to support all of them, but without considering delivery scope in the context of these activities, it is very difficult to prioritize and compare deliverables. The hierarchical nature of the map clearly shows who creates an impact and how that contributes to the goal. This clear visualization allows us to decide which impacts best contribute to the goal and identify the risks; this helps immensely with prioritization.

### 2.4. First Workshop

The first workshop was designed to consist of three parts: free writing, the hero's journey and writing poetry. The workshop was developed around the subject "Data Ethics in a Post-Covid Era" and lasted three hours in total. At this time, only the first two sections were completed, and the evaluation. Each subject was chosen with the use of the innovative tools of Trend Radar from ITONICS (Itonics-innovation.com, 2021) [24].

The workshop was held with six students of the Human–Computer Interaction MSc course of the University of Patras, their tutor and a moderator/instructor. It is usual for creative workgroups to consist of six to twelve persons (Morgan, 1998) [25], and this specific group of postgraduate engineer students was considered as a good group for an

initial assessment of the technique. Wilkinson (1998) [26] says that one should confront researchers with the pluralistic and dynamic nature of human perception; "and the fluidity, contrasts, and plurality of respondents' views, feelings, and experiences." In the beginning, all participants were introduced to the technique, the subject, the structure of the workshop and the tools Zoom and Google classrooms that were used as a means to carry it out. Later, after they passed through the Section 1, they had an introduction to free writing, and they were given fifteen minutes to practice it. When everyone finished, they were asked to read their texts to the rest of the participants—this part lasted ten minutes. The whole section lasted thirty minutes.

Evaluation of the technique was performed in three parts: The first part consisted of two open questions about the strengths and the weaknesses of the technique; the second part was a questionnaire asking students to grade the technique; and the third part was a live discussion where the students, the professor and the instructor discussed the technique.

2.4.1. Google Classrooms Description

The first page of the Google Classroom consisted of the icon for the class in Figure 1. When the users clicked on the icon "Design Ideation and creative narration," they entered the classroom, where they saw Figures 2 and 3. That page is divided into sections, and each topic consists of assignments and materials. In the Materials section, users can read relevant material about the section, and in the Assignments section, users can fill out Google Docs with the tasks that are assigned to them.

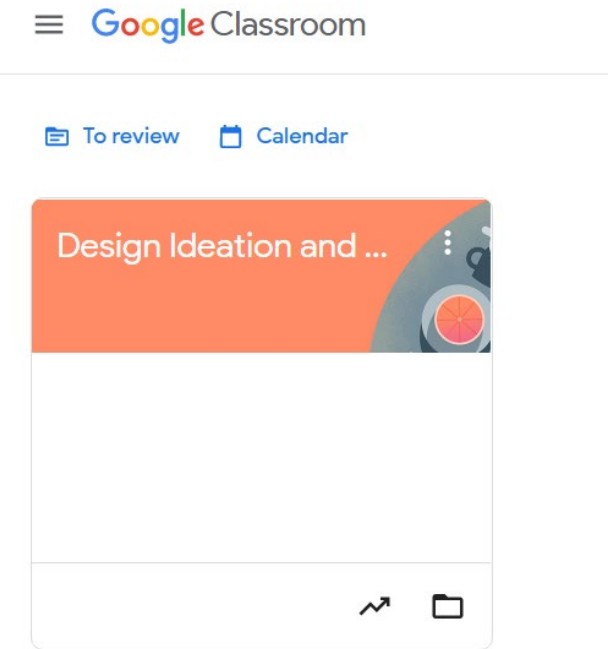

**Figure 1.** Google Classroom introductory page.

In (Figure 4) the assignment page is presented. When the participants filled out an assignment, the class administrator could find it on this page. Assignments were stored in documents.

The evaluation section consisted of two assignments where participants added their answers in plain text, and one assignment that was a questionnaire (Figure 5).

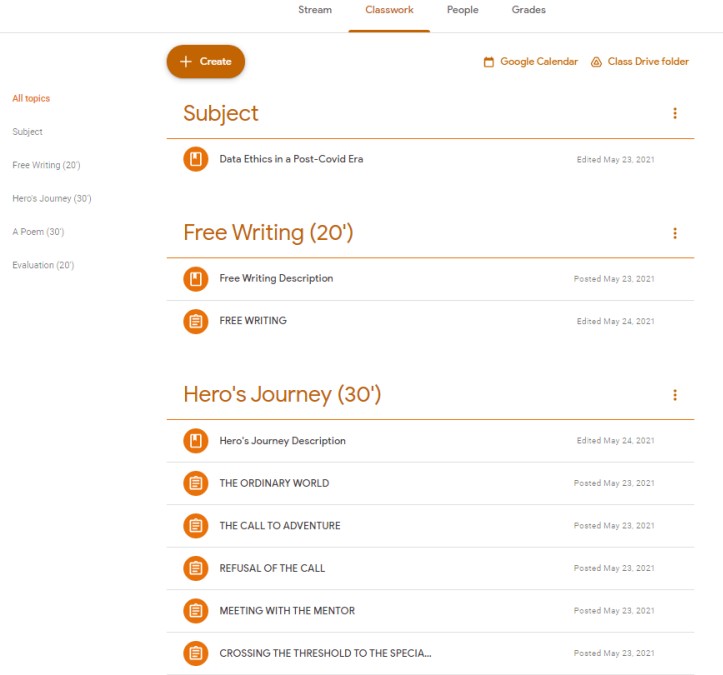

**Figure 2.** Google Classroom, classwork (first part).

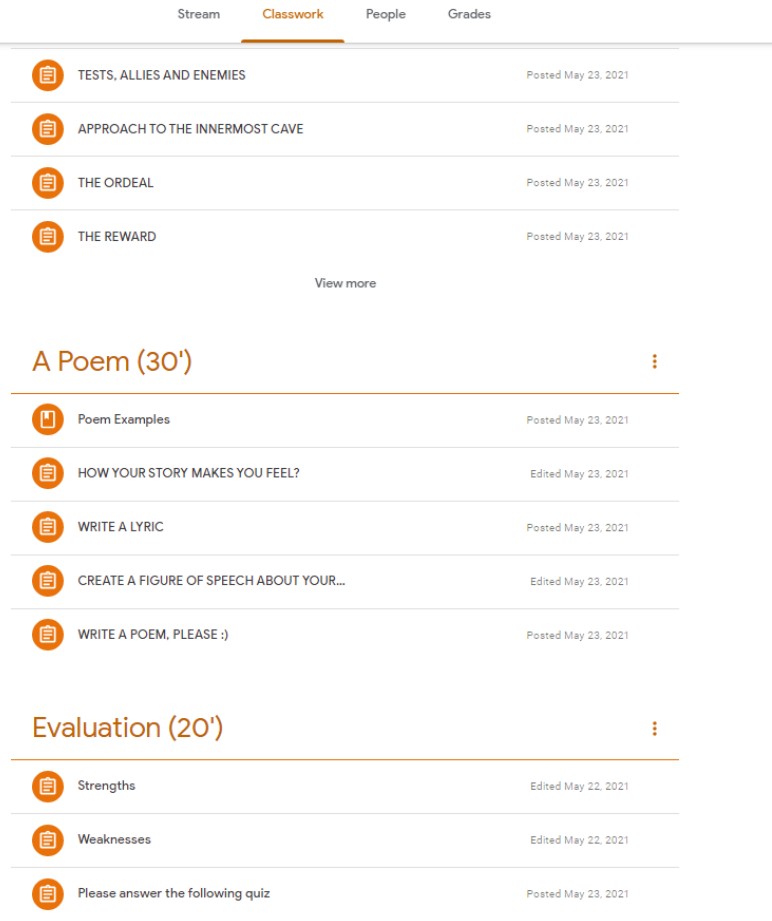

**Figure 3.** Google Classroom, classwork (second part).

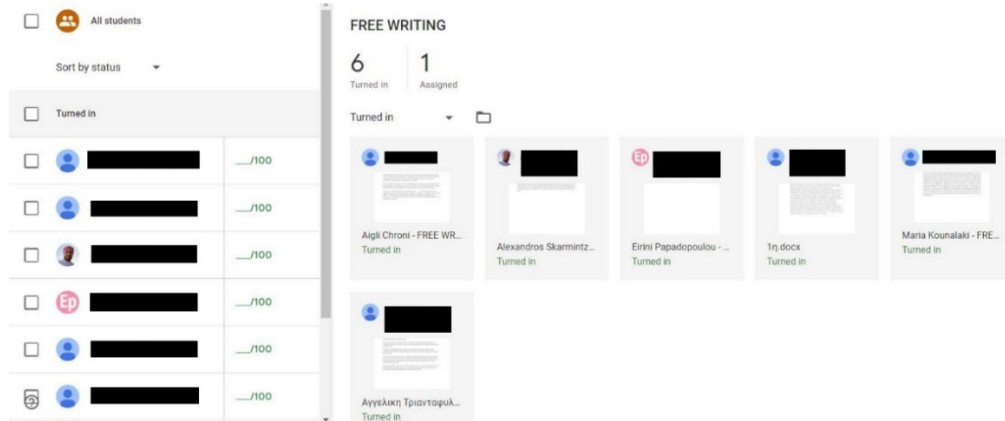

**Figure 4.** Assignment page.

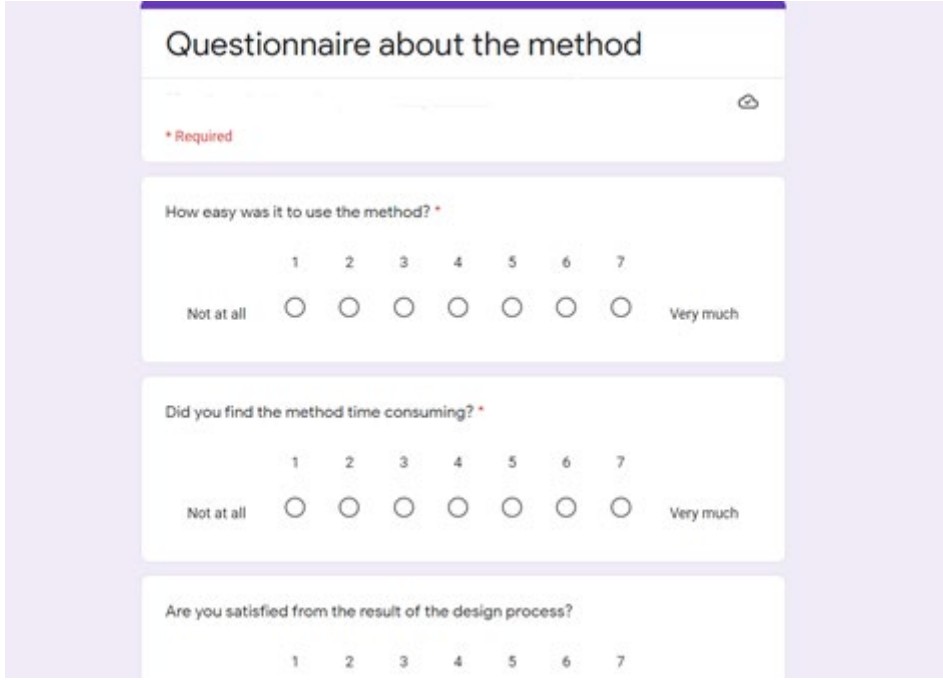

**Figure 5.** Questionnaire.

### 2.4.2. Warm-Up Stage: Free Writing

Free writing is a technique that allows the subconsciousness to reveal itself. Thus, it is very personal, and the writer should feel secure. In that part of the process, participants were informed from the beginning that they were not obliged either to share their texts or submit them. It has been, usually, used as a prewriting method in academic environments, in which the individuals write incessantly for a certain period without worrying about metaphorical apprehensions or conventions and mechanics.

As opposed to brainstorming, in which ideas are only cataloged, in free writing, the person writes sentences to make a paragraph that will include whatever appears in its mind. Dorothea Brande was an early proponent of free writing. In her book Becoming a Writer (1934) [27], she recommends to her readers to sit and write for thirty minutes every morning, as fast as they can. Peter Elbow advanced free writing in his book Writing Without Teachers (1973) [28], and it has been popularized by Julia Cameron through her book The Artist's Way (1992) [29].

The writer was to write without any concerns about grammar and spelling, and avoid making corrections. Free writing is a way to express one's most inner thoughts, unload and learn about yourself.

### 2.4.3. Main Workshop Part: Hero's Journey

When the free writing part was completed, students proceeded to the Section 2, the hero's journey. The hero's journey is a widespread pattern of stories that talks about a hero who goes on an adventure, is victorious in a decisive crisis and returns home changed or transformed.

Created by Joseph Campbell in his emblematic book The Hero with a Thousand Faces, the hero's journey or monomyth is a structure used to form a convincing plot based on myths and stories from a broad range of cultures. A monomyth is a story including a protagonist starting in an ordinary situation, traveling into a special one, dealing with tests and ordeals (obstacles) before being rewarded (goal) and ultimately returning to the ordinary situation in a changed state. The hero's journey has been modified by Christopher Vogler (Vogler, 2017) [30] into twelve different stages using clearer language and has been successfully applied to several popular movies, such as Star Wars.

When the workshop's participants finished the steps of hero's journey (Figure 1), they shared their stories with the group. Based on the stories they presented, they were asked to ideate new products. The whole section lasted eighty minutes: ten minutes for the introduction and the instructions, forty minutes for the design of the story, ten minutes for the sharing of the stories, ten minutes for the conceptions of the products and ten minutes for the sharing of the products with the participants. After the end of that session, students got a ten minute break.

At the end of the workshop, students were asked to evaluate the total process. The evaluation lasted twenty minutes, and it was followed by a ten minute discussion, where students and tutors gave feedback based on their personal experience, and notes were kept by the moderator.

### 2.4.4. Evaluation

All students pinpointed as a weakness the duration of the workshop. They claimed that they needed more time to be more relaxed and focused, and produce better results. One student wanted more information about the subject.

The students identified several strengths. They liked both of the sections, and some more specific strengths were:

- The structured form of the workshop with small, well-defined and described steps.
- The collaboration with other students.
- That they were able to produce results in terms of ideation and scenario building.
- They felt positive feelings when they followed the hero's journey process because they were able to create a scenario. One mentioned that she would identify herself with the hero protagonist and thus tried to rescue the hero.

The questionnaire consisted of five questions, and each one was marked from one to seven.

In the first one (Scheme 1) about the ease of use, the majority of the students (66.7%) voted for the technique being very easy.

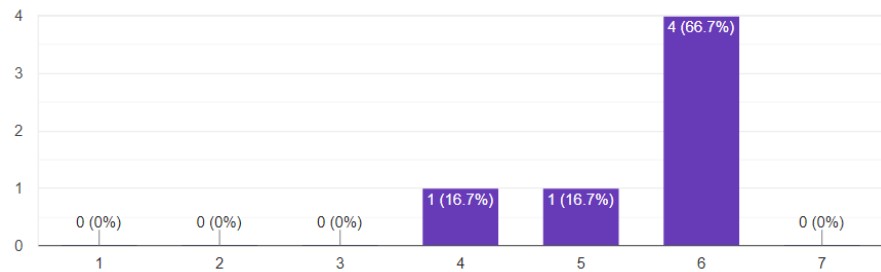

**Scheme 1.** "How easy was to use the method" question (1—not at all, 7—very).

In the second one (Scheme 2) about the time consumption of the technique, half of the students (50%) found it "not bad."

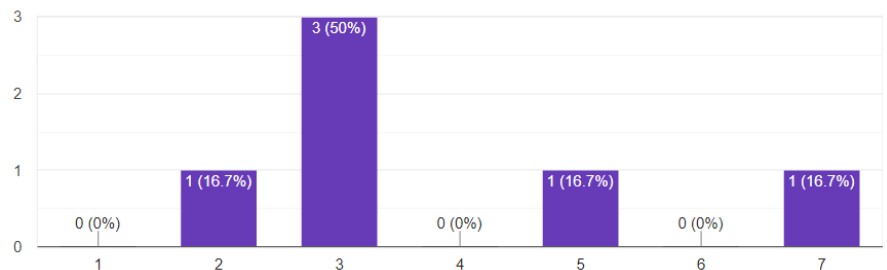

**Scheme 2.** "Did you find the method time consuming" question (1—not at all, 7—very much).

In the third one (Scheme 3) about the results of the process, most of the students (5/6) found the results very satisfying.

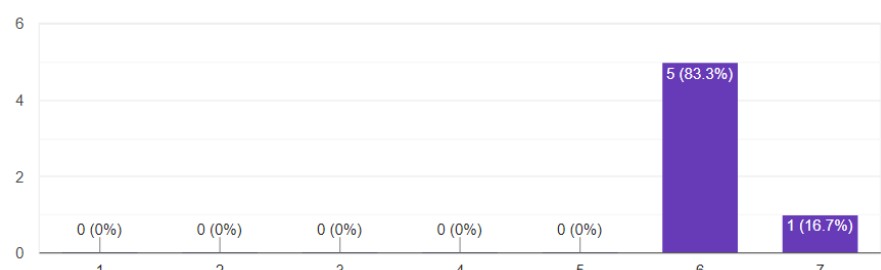

**Scheme 3.** "Are you satisfied with the results of the design process" question (1—not at all, 7—very much).

In the fourth question about the possibility to use the technique again, all of the students answered that they would use the technique again (Scheme 4).

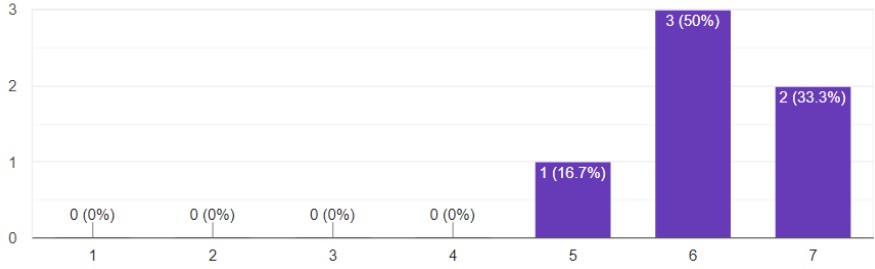

**Scheme 4.** "Would you use the method again" question (1—never again, 7—yes certainly).

In the fifth and last question (Scheme 5), all of the participants answered that they would recommend the technique to a friend (Keiningham, et al. (2007)).

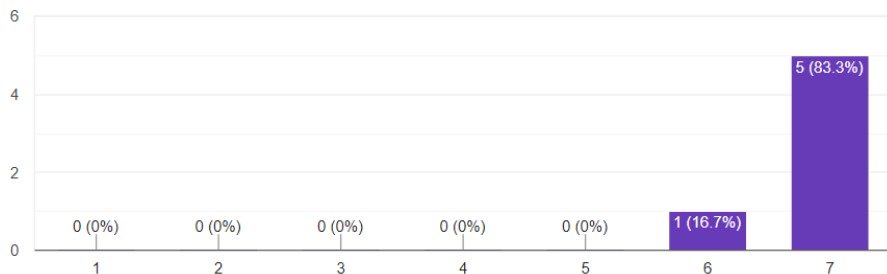

**Scheme 5.** "Would you recommend the method to a friend" question (1—never, 7—certainly).

In the third part of the evaluation, a discussion took place. At that point, the participants compared the technique with previous techniques that they had tried in class, and they found creative writing the most effective and pleasurable. Then again, they said that they needed more time. There was not any preset list of questions, and the instructor wrote down all responses. The evaluation was primarily qualitative, as there were no significant points and topics to be qualitatively researched.

### 2.5. Second Workshop

The first workshop was introductory, and it tested the audience's adoption of the technique. The results were considered extraordinary by the researchers, since everyone liked the technique, and the perceived products were very satisfying. The purpose of the second workshop was to enrich the technique by comparing it with another technique and by improving the validation methodology of the workshop. After analyzing the first workshop's results and taking into consideration all the participants' opinions about the whole process, it was decided to make some changes in order to eliminate the weaknesses and come up with even better and more accurate results. The most significant changes were addressed into three sections of the workshop. They were about the structure, the participants and the estimated time. In more detail, as almost all of the first workshop's participants claimed the time was not enough to accomplish all the tasks, the tasks' structure changed: it became stricter and participants were given more time for the more important and creative tasks, such as the story making and the products' exporting. The participants were also asked to answer the demographic characteristics section at least a day before the workshop in order not to waste unnecessary time during the main part of the workshop. Another change was in the formation of the groups. In the first workshop, the participant selection was arbitrary and they came from various fields. The second time, the participant selection was stricter. We selected people who were studying in the computer engineering fields. This happened to study how creative a group of people could be who have been taught to think or work in a strict mathematical framework. After that, the selected people were separated into three classrooms, this time in order to work by using three different methods.

### 2.5.1. Demographic Characteristics

The second workshop was divided into two parts and three classrooms. In the first part, students gave their demographic characteristics. According to their characteristics, they were divided into homogenous groups. One of the major criteria was students' creativity. It is impossible to compare the results of a creativity workshop if the creativity of the participant groups is not calibrated. Divergent thinking performance was evaluated with the use of the alternative uses task (Benedek, Könen, & Neubauer, 2012) [31], in which, by using a screen, we displayed to all the participants two ordinary items, for which, within five minutes they had to observe and note down any possible realistic use they could figure

out. Another criterion was the English language. Therefore, the three different classrooms were formed based on the answers given in the demographic characteristics questionnaire.

2.5.2. Main Part

The techniques that were applied were impact mapping, creative narration and a combination of these two. In the second workshop, the impact mapping was examined as an addition to the creative narration technique. In the first classroom, students followed the steps of the creative narration exactly as in the first workshop. In the second classroom, impact mapping was added to the steps of the hero's journey, and in the third classroom, students followed the steps of impact mapping. In all three classrooms were some similar steps at the beginning, such as the introduction, the "keep notes" part that was optional for each participant and introducing the topic. When the workshop was over, they were asked to write down the method's strengths and the weaknesses. The introductory section was actually about meeting the instructor and getting an idea about the workshop. Afterward, the "keep notes" section was not only one step, as it lasted through the whole workshop, and one member of each team had to keep notes about the thoughts of all of the team members. Following that explanation, participants were given the topic of their story. Off course, also a common step in all classrooms is the "products" section, which was the main purpose of the whole workshop. At this point, each of the three classrooms had to find as a group some innovative products or services that emerged from their stories. Each description of the product or the service they thought about had to include a general description of it, its functions, the interactions it would have with the user and how they would experience it.

Firstly, the classroom that applied impact mapping is analyzed. Impact mapping has the following steps (Figure 6).

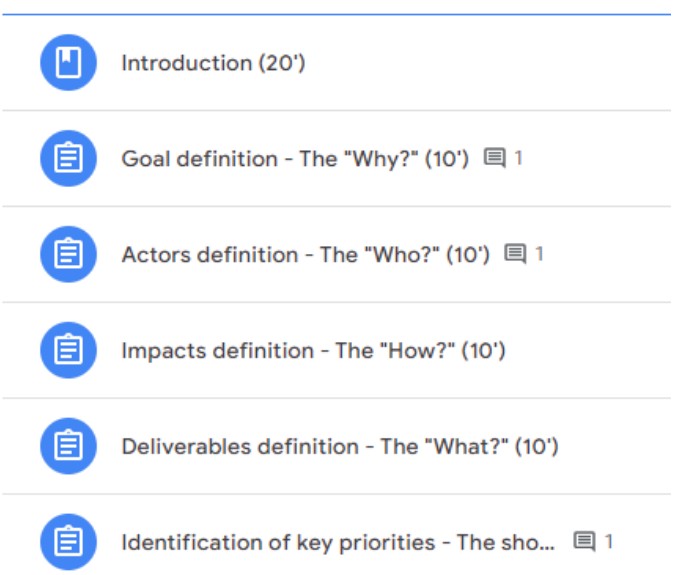

**Figure 6.** Impact mapping steps.

As you can see in the figure above, the estimated time for impact mapping was about 70 min, and each step took about 10 min. In more detail, participants in the first place came in touch with the theory of impact mapping: what it is about, who should use it and why one should use it. Teams used impact maps to discuss assumptions, align with organizational objectives and develop greater focus in their projects by delivering only the things that led directly to achieving organizational objectives. This also reduced

extraneous activities. The participants were asked to find some products or services using this technique. In the first step, participants had to address their goals, and why they were doing the test. The purpose of goal definition is to allow the delivery organization and business sponsors to re-evaluate the plan as new information becomes available. For this reason, goals tend to be **SMART**:

**S**pecific,

**M**easurable,

**A**ction-oriented,

**R**ealistic,

**T**imely.

Goals should not be about building products or delivering the project's scope. They should explain why such a thing would be useful. Goals should also present the problem to be solved, not the solution. Participants have to keep in mind that they should avoid design constraints in a goal definition.

In the second step, they were asked to pinpoint the "actors" who would define the outcome. They had to be specific and avoid generic terms such as "users"—different categories of users might have different needs, and not all users of a product will be important to consider. There are three main types of actors: the primary actors who have fulfilled their goals, secondary actors who provide services and off-stage actors who have an interest in the behaviors but are not directly benefiting or providing a service.

The second level of an impact map views the actors from the perspective of the goal. At this part, they had to write down only the impacts that help move in the right direction towards the central goal, while avoiding talking about software ideas. They also had to think about any changes in actors' behavior and their negative or positive impacts. In other words, they had to note down the impacts for each of the actors and connect each impact with the corresponding actor.

In the third part, they were asked to define their scope. They had to think about what can happen reach the goals they wanted to achieve.

In the final stage, the participants had to find the shortest path on the map to reach their goals. For achieving that, they had to make sure their deliverables would fulfill some conditions, such as being realistic and valid.

The second classroom worked based on both creative narration and impact mapping. The section above displays impact mapping's steps and method. For the second classroom, those steps were incorporated inside the creation of the story (Figure 7).

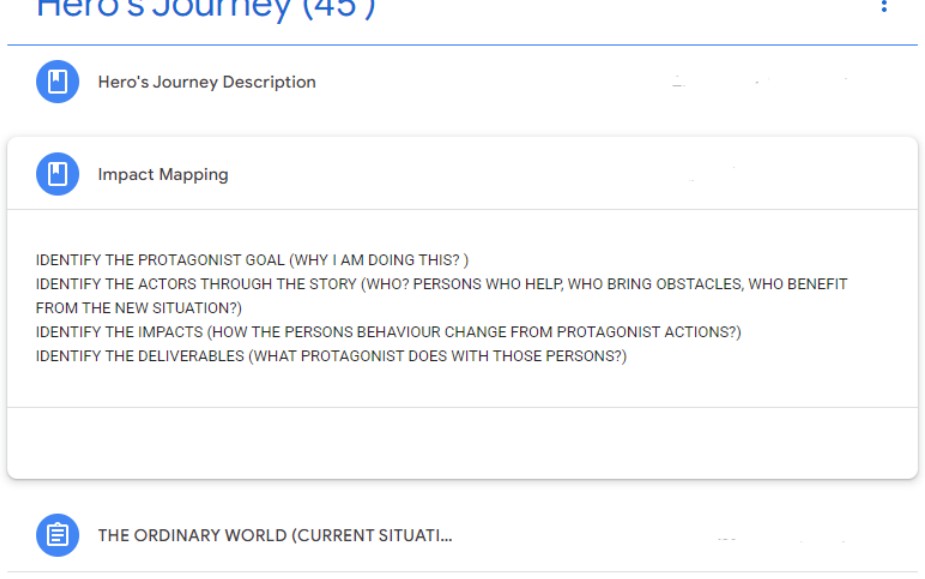

**Figure 7.** Creative narration and impact mapping.

### 2.5.3. Keep notes and Focus Group

When a workshop is conducted, there is the need to gather, analyze and compare its results. However, it is difficult and time-consuming to interview each participant separately. In order to make this process more effective time-wise, the "focus group" technique has been often applied. The number of people who constitute a focus group is extremely important: the groups will not be as effective and accurate as expected if they are large, but opinions differ on optimal sizes. Some suggest between 8 and 12 people (Robson, 2002), while others argue that smaller groups of 5 to 7 participants might be more appropriate for an in-depth conversation (Krueger, 1994). It has also been recommended to rely on more than one focus group because a single focus group can be impassive or unreliable. Being able to work with two or even three focus groups significantly raises the chances of accurate outcomes and success (Krueger, 1994). Furthermore, a mixture of different personalities in one group will inevitably bring up a wide variety of viewpoints and insights. Through their interaction, they will come to agreements and disagreements that up to a point are desirable, because varying viewpoints can lead to a broader understanding. These conflicting perspectives might also lead one to new areas for further study (Brown, 1999) [32].

Focus groups are mostly semistructured or unstructured. In the case of a fully structured focus group, it means there are questions in a specific order to ask each of the participants, and there is no real interaction between them. A fully structured focus group would fundamentally be equal to multiple individual interviews conducted instantaneously.

Applying the method of "focus groups" can also provide some challenges, such as taking more time than expected because of excessive discussions or asking less than the planned questions because of a lack of time. Another critical issue is the possibility of going into sensitive topics that some participants may not be willing to discuss in front of others, such as personal affairs or their finances. When conducting a focus group, one must be careful to avoid power struggles or other confrontations with participants, as such battles can sabotage the whole process (Brown, 1999) [32].

Extracting accurate information through research based on focus groups requires skillful facilitation. In detail, it requires managing personalities, encouraging participation from all participants, keeping the conversation going, monitoring the clock and working through a list of questions, all while collecting the statistics that are the crux of the whole effort. Depending on the people who participate in the groups, this process can be quite challenging, especially if there is only one moderator, but most of the time there are two of them. These collaborators can work together to ensure successful data collection.

The selection of focus group participants can be an art in itself. First, it presupposes some thought about the groups' synthesis, such as if they are going to be homogenous or inhomogeneous, and if the participants know each other or work together and how that will affect the group's dynamics. Homogenous groups have the disadvantage of narrowing the range of perspectives. On the other hand, a group composed of various personalities may bring about some problems, such as difficulty in finding common ground between the members (Krueger, 1994). In any case, participants in focus groups should have an interest in the topic, and they should be willing to participate positively (Brown, 1999) [32].

In both focus group and keep notes sections, participants stated that they were not familiar with the free writing part, and they were uncomfortable for the first few minutes. Afterward, they enjoyed it and found it a good warm-up exercise.

Something else that was found in the keep notes part was that students needed collaboration with other students. With collaboration, the process was more fun, and students complemented each other, especially students with different knowledge backgrounds.

In the end, every team stated that they liked the process and that they might use it again in the future.

### 2.5.4. Products

To compare the three different techniques, it is necessary to compare the conceptual designs of the products and the services produced by the process. The metrics for measuring ideation effectiveness (Shah, Smith, & Vargas-Hernandez, 2003) [33] were novelty, variety and quality. The evaluation was designed to be accomplished by experts in the fields of design, together with experts in the contexts of the workshop.

The products of the first group, the one in the creative narration classroom, were much better than those of the other classrooms. Those products were seven in number, whereas the product of the second classroom, using creative narration and impact mapping, was singular, and in the third classroom, using impact mapping, the same. Therefore, the first classroom was considered much more successful.

### 2.5.5. Evaluation

After analyzing the first workshop's results and the participants' thoughts and opinions about the whole process, the evaluation of the second workshop was different. We used a questionnaire inspired and completely written by the author for the first workshop. Although it gave plenty of interesting results, for the evaluation of the second one, it was decided to apply an approved one known as the "User Experience Questionnaire." The User Experience Questionnaire (UEQ) is a fast and reliable questionnaire to measure the user experience of interactive products. It is available in more than thirty languages, it was used in English throughout the whole workshop and it is easy to use due to rich supplementary material. It has a default and a short version. For the aims of this research, the short version was used. It contains seven different questions with answers scaled from 1 to 7 (obstructive—supportive, complicated—easy, inefficient—efficient, confusing—clear, boring—exciting, not interesting—interesting, conventional—inventive, usual—leading edge) (Figure 8).

**Figure 8.** Sample of the User Experience Questionnaire.

Each group answered those questions separately, so that it would be practical to compare the results. Indicative answers of the group "A. Creative narration" are displayed below.

First is the "obstructive-supportive" question (Scheme 6).

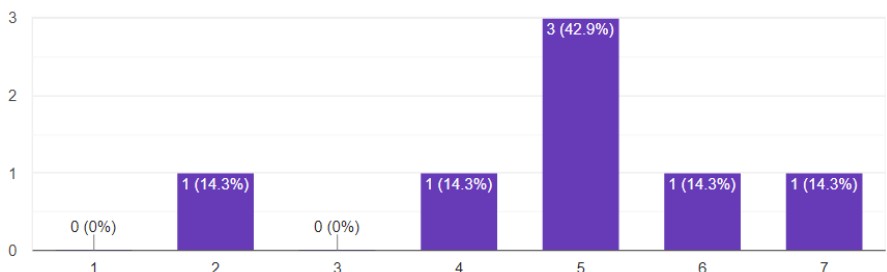

**Scheme 6.** Obstructive—supportive question of the User Experience Questionnaire.

Second is the "complicated-easy" question (Scheme 7).

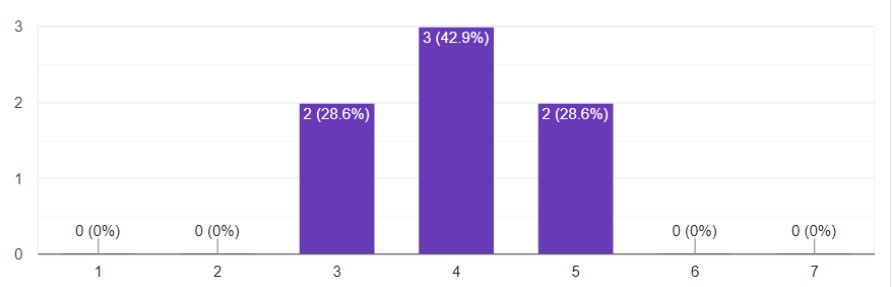

**Scheme 7.** Complicated—easy question of the User Experience Questionnaire.

Third is the "inefficient—efficient" question (Scheme 8).

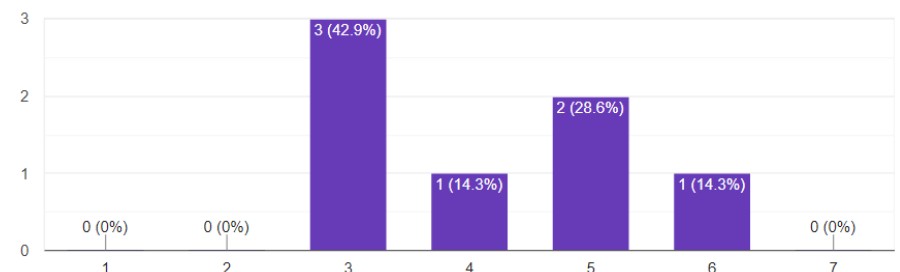

**Scheme 8.** Inefficient—efficient question of the User Experience Questionnaire.

Fourth is the "confusing-clear" question (Scheme 9).

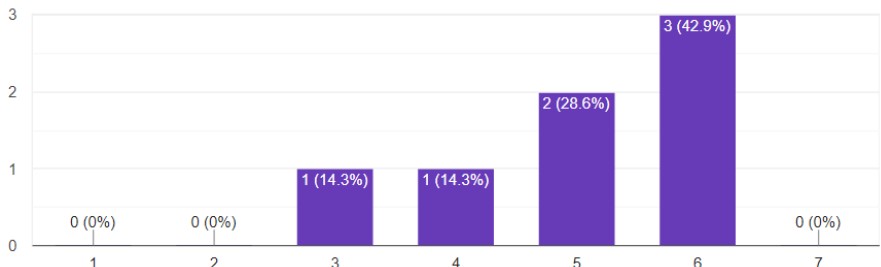

**Scheme 9.** Confusing—clear question of the User Experience Questionnaire.

Fifth is the "boring—exciting" question (Scheme 10).

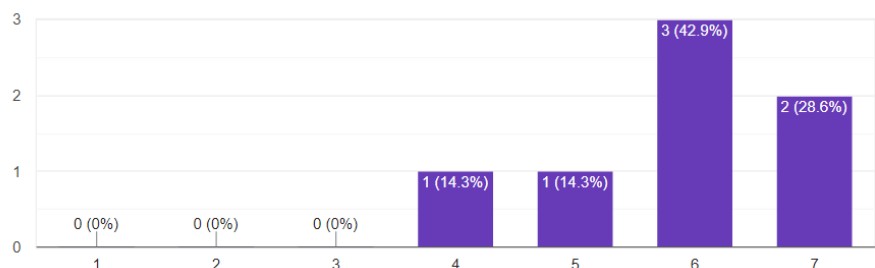

**Scheme 10.** Boring—exciting question of the User Experience Questionnaire.

Sixth is the "not interesting—interesting" question (Scheme 11).

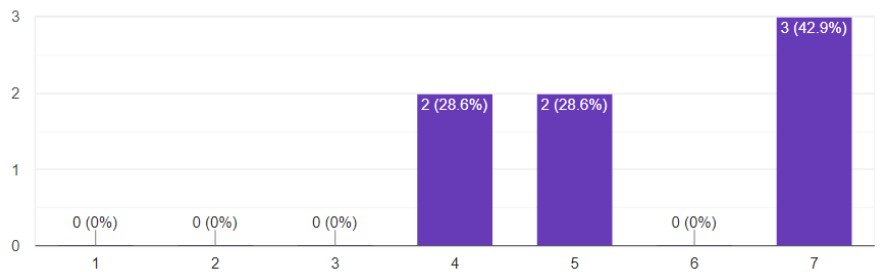

**Scheme 11.** Not interesting—interesting question of the User Experience Questionnaire.

Seventh is the "conventional—inventive" question (Scheme 12).

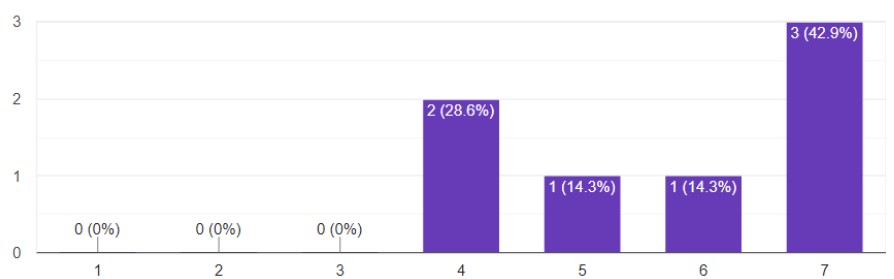

**Scheme 12.** Conventional—inventive question of the User Experience Questionnaire.

Eighth is the "usual-leading" question (Scheme 13).

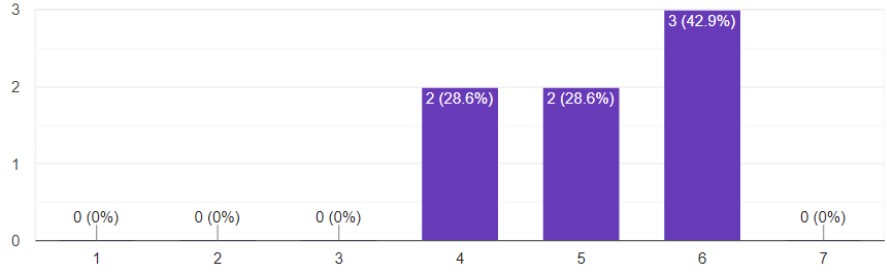

**Scheme 13.** Usual—leading edge question of the User Experience Questionnaire.

The "User Experience Questionnaire" provided the opportunity to measure the statistics using the terms "pragmatic quality" and "hedonic quality." While the pragmatic qualities refer to the perceived usefulness, efficiency and ease of use (so-called utility and usability aspects), the hedonic qualities take into account the "joy of use" and emphasize stimulation, identification and evocation generated by the use of a system or a product.

The results that came from the "User experience Questionnaire" are presented above, and were categorized by group with the use of the data analysis tools on Excel files provided on the UEQ website.

As is shown in Scheme 14, the team which applied creative narration technique seemed to be satisfied overall. However, when it comes to the pragmatic quality measurement, it looks like they had several difficulties to overcome, but the hedonic quality measurement shows that they really enjoyed the method.

| Scale | Mean | Comparisson to benchmark | Interpretation |
|---|---|---|---|
| Pragmatic Quality | 0.5 | Bad | In the range of the 25% worst results |
| Hedonic Quality | 1.571428571 | Good | 10% of results better, 75% of results worse |
| Overall | 1.04 | Above Average | 25% of results better, 50% of results worse |

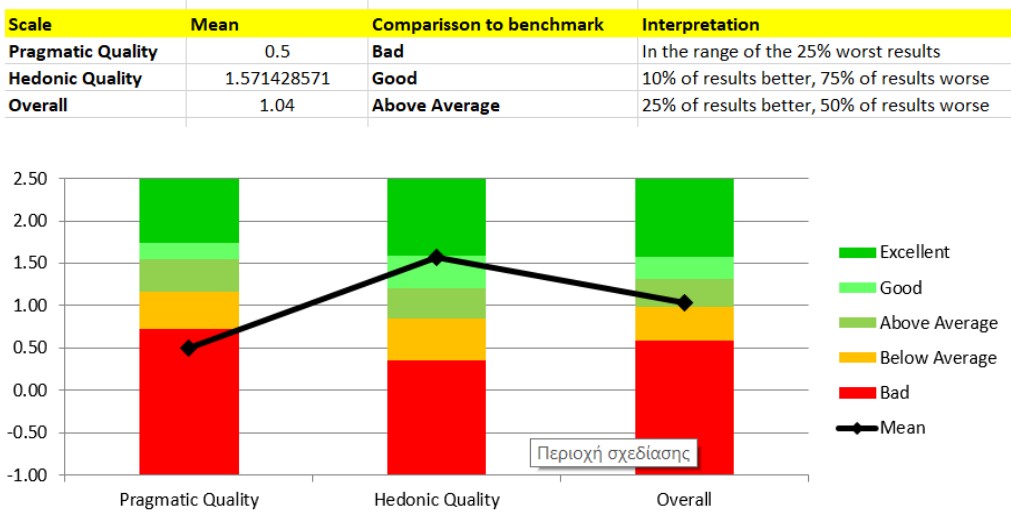

**Scheme 14.** Creative narration's UEQ results.

The second team that applied the impact mapping technique did not seem as pleased as the first group (Scheme 15). Both the pragmatic and the hedonic qualities show us that it was a difficult method for them when it came to its efficiency and ease of use, and they did not enjoy the method—probably because they found it complicated, but that is an assumption, not a proven conclusion.

| Scale | Mean | Comparisson to benchmark | Interpretation |
|---|---|---|---|
| Pragmatic Quality | -0.8125 | Bad | In the range of the 25% worst results |
| Hedonic Quality | -0.125 | Bad | In the range of the 25% worst results |
| Overall | -0.47 | Bad | In the range of the 25% worst results |

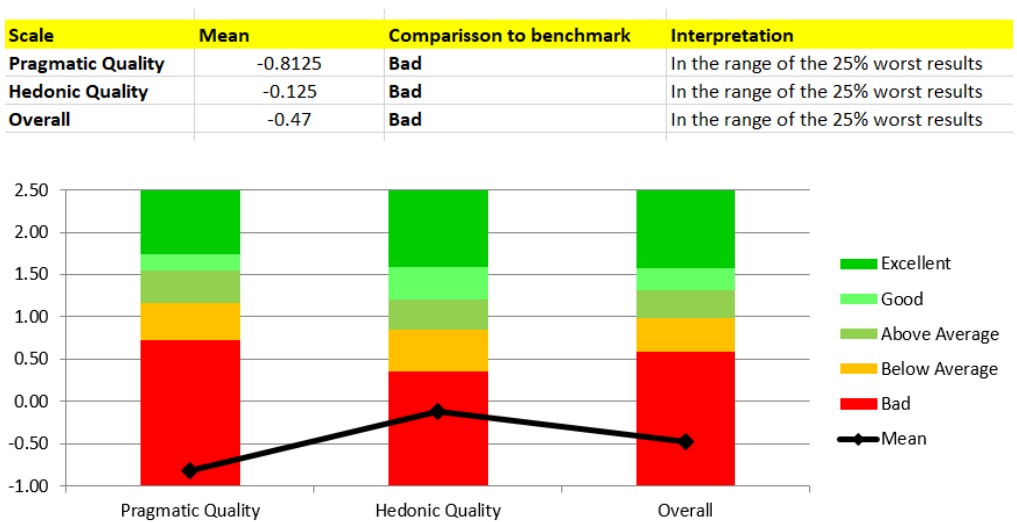

**Scheme 15.** Impact mapping's UEQ results.

The results of the third team (Scheme 16) which used both of the techniques show that the participants enjoyed this method, but they still reported some negative traits, such as inefficiency and difficulty.

| Scale | Mean | Comparisson to benchmark | Interpretation |
|---|---|---|---|
| Pragmatic Quality | 0.375 | Bad | In the range of the 25% worst results |
| Hedonic Quality | 1.5 | Good | 10% of results better, 75% of results worse |
| Overall | 0.94 | Below Average | 50% of results better, 25% of results worse |

**Scheme 16.** Impact mapping and creative narration's UEQ results.

Overall, it can be certainly said that the group which enjoyed the workshop more and was the one which faced fewer problems was the one that worked with the method of creative narration.

In addition to the "User Experience Questionnaire," all participants were also asked to write down, as individuals and not as a group, the strengths and the weaknesses they could see in the workshop overall, and more specifically regarding the method they applied. In order to come up with useful results, their answers were examined and compared group by group (Table 1).

**Table 1.** The three techniques of the second workshop.

| | |
|---|---|
| Creative narration | They needed time to get familiar with the process but eventually, they enjoyed it and found it effective in terms of finding new products |
| Creative narration & Impact Mapping | They needed time to get familiar with the process and to find the subject of the story but eventually, they enjoyed it and found it effective in terms of finding new products. They skipped the impact mapping part. |
| Impact Mapping | Students did not follow the process and tried from the beginning to find the idea. |

First, we will address the group that worked based on creative narration. This group's participants had a lot to say about the positives of "free writing," since they found it extremely interesting. Most of them found it useful and cheerful working in groups, as this, as they said, helps with creating a collaborative environment. Furthermore, they found this method interesting because it is a more creative way of working than how they had been taught to work as engineers. They also referred to it as a new way of finding ideas and solutions naturally. As a weakness, they claimed they needed more time to do all the tasks correctly. One said that he would like less time for the "free writing" in order to have more time for the story and the final products. Another one alleged that the environment led him to create a product that would not work to the real world.

The second group, which had to work based on impact mapping, is the one that worked through zoom, and that caused several problems, such as poor communication between them. From the beginning, it was not clear to them what to do and why. They also did not understand, and they wanted the impact mapping technique, with which they were not familiar, explained before the workshop. Overall, they liked the workshop's structure, they found it interesting and they would love to use it again in the future.

The third and last group applied both impact mapping and creative narration. The strength that was mentioned from most of the participants was that they enjoined working in groups because they were not used to it and it gave them the chance to collaborate in order to achieve a common goal. The majority did not face any difficulties. On the other hand, a small number of participants mentioned that they had some difficulties working in groups because they had not worked with this method, nor were they familiar with accepting other people's ideas. They also claimed that they were amused by the fact that "there was actually something that could be thought and created."

As a final evaluation, after thinking of the opinions of all the three classrooms, what came up is that, if not all of the participants, most of them as engineers were not well acquainted with techniques based on their creativity. They were surprised when they saw that they could really come up with decent results after this process. As expected, some appreciated the teamwork and their interactions with others, and some faced problems, especially those who worked online. To sum up, a common strength to all teams was the cultivation of their creativity, and the most common weakness was the lack of time or its management.

## 3. Results and Discussion

### 3.1. First Workshop's Results

The first workshop aimed to improve and validate the creative narration technique. It also focused on producing some preliminary results about the usability and the user experience of the method.

Overall, the participants reported feeling satisfied with their outcomes and with the process steps they had to follow. They reported feeling free and able to immerse themselves in the creative process. One weakness reported was the need for more time to produce more elaborated results.

The participants enjoyed spending time on this exercise, and their collaboration was flawless. The use of the online tools did not bring any delays, and no one mentioned the lack of personal contact as a drawback. The structure of the workshop with small, well-described steps helped the students who participated to produce results and decode the obscure scenario building process into a fun and enjoyable game-like experience, since the students were able to collaborate well.

Apart from the evaluation results, proof of the success is that one of the participants, who is a teacher at a primary school, asked to use the technique in her classroom.

### 3.2. Second Workshop's Results

After taking into consideration what the participants wrote in the sections of "keep notes," "strengths," "weaknesses" and "how was your experience? (UEQ)," and the products they came up with, we were able to conclude on some outcomes of the whole process of the second workshop. First, even if it is obvious that the needed changes worked out, a lack of time was still reported by some participants, which was also the cause for some of them having trouble understanding either the main purpose or what to do and why. Probably due to the nature of the participants, who were studying computer engineering, in the beginning they faced difficulties when they were asked to freely write down their thoughts. However, as it turned out, all they needed was a few minutes to get familiar with the whole process and start feeling free to express themselves.

When it comes to the preferred technique, we received clues that showed us which one they enjoyed more and seemed easier for them. We consider indicative the fact that the team which followed the creative narration technique was the one that came up with the most products. They found seven products, whereas the other teams found one each. However, that is an assumption based on only one piece of evidence.

Concerning the UEQ, the method of creative narration and the one that combined both impact mapping and creative narration had about the same effectiveness to the participants, unlike the impact mapping, which seemed the worst.

*3.3. Discussion*

The first workshop presented in this paper was the first step in a series of workshops that aim to improve and validate the creative narration technique. We recognized and analyzed all the assets and the problems of the process in order to correct them for the accomplishment of the second workshop. Furthermore, the second workshop was guided and applied in the context of students for educative purposes. The second workshop's participants were studying in fields related to computer engineering, contrary to the first one, in which the professions of the participants had greater variety.

Since the users' experience and the results of the technique were satisfactory, the second workshop aimed to enrich and calibrate the creative narration technique. Thus, a new section was added where the participant had to write a story for a protagonist or another character when creating the products. As was proven in the results section, the participants enjoyed this technique, which was also proven effective as long as the team came up with decent and satisfactory products. Therefore, there is still work to be done on the workshop's time management. Even though it was better in the second workshop, there were still complaints about not having enough time. Despite that, they were pleased with the workshop's structure, and those who had problems understanding some tasks accused the lack of time. An idea for the upcoming workshops is to separate them into two parts in order to have enough time.

Another approach that is being focused on is enhancing the functionality of the technique. It will be tested if the technique can be used for trend forecasting.

## 4. Conclusions

The research reported here started with an introduction to creative narration, a solid stepwise technique that brings in ideation elements from creative writing practice. However, scenario-based ideation does not have a clear structure to define the setting and characters, (although personas are sometimes used in this respect, but in a different context). The reported technique tries to remedy that shortcoming of the existing practice of scenario-based ideation. Six to twelve participants are considered ideal for contained creative workshops. The participants here were engineers (computer and electronic engineers mainly, which are seen as appropriate trades for the 4th industrial revolution, as described in the Introduction). We assessed the technique proposed here (with a hero's journey as its main point) in a group of six. It seems to bring together elements of the foreseen problem space, with more clarity. It assists, due to its stepwise approach, the groups of engineers to have a structured creative process and exchange ideas. Engineers—unlike designers or architects who are trained heavily for creative ideation and can handle more vagueness when they exchange creative ideas—are not usually trained in ideation techniques. Overall, the prospects of this technique, as an ideation tool for assisting engineers in the creative workshop process, seem promising.

All in all, writing is a difficult procedure, especially for people who are not familiar with it. Future experiments will examine ways to enrich the technique, not only to make it more efficient and more pleasurable for the participants, but also to add more functionalities.

**Author Contributions:** Conceptualization, D.M.; Methodology, D.M.; Project administration, D.M.; Supervision, I.M.; Writing—original draft, D.M. All authors have read and agreed to the published version of the manuscript.

**Funding:** Distant MMLAB, Hellenic Open University, as well as Erasmus training mobility (travel regarding academic collaboration and workshop evaluations).

**Institutional Review Board Statement:** Not applicable.

**Informed Consent Statement:** Informed consent was obtained from all subjects involved in the study.

**Data Availability Statement:** Not applicable.

**Conflicts of Interest:** The authors declare no conflict of interest.

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
