# Peer review of "Creative Narration as a Design Technique"

_information, doi:10.3390/info13060266_

Round 1

Reviewer 1 Report

In this paper, the authors present two case studies where technique ‘creative narration’ as design technique was used in the contexts of two creative workshops. However, this technique is well-know. Although this paper seems interesting I have some comments.

In my opinion, this paper should be improved including more current references (preferably JCR references) with previous experiences from other authors about this topic. Likewise, this state-of-the-art should include the main findings and drawbacks from related works and, therefore, it would be innovative.

The structure of this paper is appropriate. However, I recommend to merge Sections 3 & 4 (Results and Discussion)

Regarding formal aspects:

  1. Please, arrange figures, charts and tables throughout the manuscript.
  2. Please, improve the quality of charts (1 to 5) (more resolution and better definition).
  3. Please, check the grammar and spelling of the English.

In general, more effort is required.

Author Response

The article underwent a major revision .

Furthermore,

  • the sections 3&4 were merged,
  • the figures, charts, and tables were rearranged,
  • the charts 1-5 were replaced,
  • english was improved.

Due to lack of time there were only some minor improvements to the theoritical part. 

Reviewer 2 Report

This paper reports creative workshop by merging many existing techniques.

First, this paper is out scope of MDPI “Information” journal, which focuses on novel and cutting-edge scientific research dealing with applications that leverage information technologies. This research only used the existing technology, such as Zoom, and no finding or lessons on information technologies.

Second, they refer creative technique, but they lack scientific soundness or no systematical explanation on design studies, creative design, and creativity research. To overcome this problem, they should survey and understand many papers related to the research topic.

Third, the paper structure is very strange to read it as a scientific paper.

Author Response

The article underwent a major revision .

Furthermore,

  • the sections 3&4 were merged,
  • the figures, charts, and tables were rearranged,
  • the charts 1-5 were replaced,
  • english was improved.
  • structure improved
  • grammatical mistakes corrected 
  • punctuation corrected

Paper structure tries to follow the given template. Theoretical part needs some improvement.

Reviewer 3 Report

Review 

The work presents an interesting, extensive study on usage of Creative Narration technique and brings the scenario based ideation into practice. Generally, the study is well designed and interpreted, can arouse the public interest, and can be considered for publication. Nevertheless, a minor revision is required. 

Remarks:

  1. The text requires linguistical correction.  

The manuscript's structure should be more fluent, and such expressions like “therefore”, “moreover”, etc. would be helpful – now it is uneasy to read the separate sentences when this kind of adverbcs, consisting the inherent part of scientifical language, provides the fluency. Moreover, some sentences are too long and could be divided into shorter ones.

Furthermore, some grammatical mistakes were found:

L. 22 “enable them to be capable to adapt” should be “capable of adapting”

l. 35 “that’s” is informal should be” that has’ 

Line 41 “of separating” instead of “of separate”:

There are many problems with punctuation - in almost every paragraph, some commas are missing. 

There is many more linguistic mistakes, but the correction should be made by a professional corrector, not by the Reviewer, who is not substantially prepared to make such action. 

2. Line 59-60 “Scientists have proved that the theory of action is more close to reality” – please provide the reference.

3. Line 32 and line 33 “The produced scenarios” Which scenarios? Please, specify, and describe the case more comprehensively, yet as it is not understandable for the reader who does not have access to the original article.  

4. Line 52 “be implemented by the engineers”? Please, correct the sentence

5. Figures 8 and Charts 6-1, as well, as the following tables are not consistent in the matter of size and the font with the previous ones presented in the manuscript

Author Response

The paper underwent a major revision.

Furthermore,

  • English was improved
  • Structure improved according to the remarks
  • Grammatical mistakes corrected according to the remarks
  • Punctuation corrected
  • Charts and figures corrected

Round 2

Reviewer 1 Report

This revised version has been improved according to my comments. Therefore, I recommend the publication in current form.

Author Response

Thank you very much for your review. I made some more corrections.

Reviewer 2 Report

This paper reports creative workshop  to consider "creative narrative.” However, there are many researches to consider "story telling”, similar to the “creative narrative" in a design workshops. The description part of their “creative narrative” seams the 2.2.2, 2.2.4. and maybe 2.4.3.

Although, the design thinking process add the “emphasize” stage and the “prototype” stage, but they ignore these two stage or no description how to relate their ideas to the design thinking process. In addition, there are similar problem in the context of creative problem solving process. They add “Impact Mapping” technique to the 2nd workshop but the technique is not relate to their main concept “creative narrative.” This part makes too difficult to understand their research concept “creative narrative."

They refer creative technique (most references are not academic research papers) , sometimes they lack scientific soundness. For example, they refer NGT but they use brainstorming in the workshop although NGT reject the group effect of brainstorming in social psychological research. This paper lacks systematical explanation on design studies, creative design, and creativity research. Therefore, the description lacks logic and story of their academic contribution. To overcome this problem, they should survey and understand many papers related to the research topic and then rewrite whole part of the paper. 

The paper structure is very bad to read it as a scientific paper. Proposed method, workshop, evaluation method, and results (evidence data) are merged into one chapter. Therefore, it is too difficult to read their research contribution. And their result and discussion is very inadequate: e.g. one participant opinion is used for the proof of effectiveness.

To overcome these problems, they are required to understand creativity research and design research, and then rewrite this paper following the basic academic writing style. This will be equal to resubmission, so the paper is judged as “reject.”

Author Response

Thank you for your comments. I understand that you need some more explanation on theory and the research decisions. I do not like the paper structure either, but I tried to follow the template that I was given. I fixed the NGT part and I made some more minor corrections.

Reviewer 3 Report

The manuscript was corrected and now is

prepeared for the publication. 

Author Response

Thank you very much for your review. I made some more minor corrections.
